# Correlations between Microscale Indentation Creep and Macroscale Tensile Creep of Polymers

**DOI:** 10.3390/ma16020834

**Published:** 2023-01-15

**Authors:** Miroslav Slouf, Milos Steinhart, Pavel Nemecek, Veronika Gajdosova, Jiri Hodan

**Affiliations:** Institute of Macromolecular Chemistry of the Czech Academy of Sciences, Heyrovsky Sq. 2, 16206 Prague, Czech Republic

**Keywords:** microindentation, indentation creep, tensile creep, polymers, viscoelasticity

## Abstract

We compared the results of various microscale indentation creep (microcreep) measurements with macroscale tensile creep (macrocreep) measurements of three common polymers: high-density polyethylene (PE), polypropylene (PP), and polystyrene (PS). The main objective was to verify if the short-term microcreep experiments could predict long-term macrocreep behavior of the selected polymers, whose properties ranged from very soft and ductile (PE) to very hard and brittle (PS). The second objective was to compare several creep predictive schemes: the empirical power law model (PL) and several types of phenomenological elasto-visco-plastic models (EVP). In order to facilitate this task, we developed a universal program package named MCREEP, which fits PL and EVP models to both tensile and indentation creep data. All experimental results and theoretical predictions documented that: (i) regardless of the creep experiment type, both micro- and macrocreep resistance increased in the following order: PE < PP < PS, (ii) the short-term microcreep experiments could be used to predict qualitatively the long-term macrocreep behavior, and (iii) the simple empirical power law model yielded better predictions of long-term creep behavior than the more sophisticated elasto-visco-plastic models.

## 1. Introduction

In materials science, *creep* (sometimes called *cold flow*) is a slow deformation of a solid material subjected to a long-term load. It is especially important for soft polymer materials (such as synthetic polymers and biopolymers), which typically exhibit much higher creep than hard inorganic materials (such as metals, alloys, ceramics, minerals and rocks) [1]. While the inorganic materials at common loads tend to exhibit relatively low deformations that are mostly elastic (i.e., time-independent and reversible), the polymer materials at comparable loads usually exhibit substantially higher deformations that are typically elasto-visco-plastic (i.e., time-dependent and not always fully reversible) [1,2].

Typical macroscale creep experiments (macrocreep) on polymers are performed in tension (tensile creep; ISO 899), at relatively low stresses (below the yield stress, *Y*), and the experiment times are long (ranging from hours to days or even weeks) [3,4,5]. Microscale creep experiments (microcreep) are less common, but they can be performed easily by means of modern microindentation or nanoindentation hardness testers (indentation creep; ISO 14577). In comparison with macrocreep, the maximum stresses during the microcreep indentation measurements are usually higher (above *Y*), the experiments are shorter (measurement times from tens of seconds to minutes), and the specimens can be much smaller (the typical indent size on a polymer surface is 10–100 μm) [6,7]. More detailed explanation and comparison of macrocreep and microcreep experiments is given below in Section 2, but even the basic pieces information within this paragraph indicate that microcreep experiments show two important advantages: (i) significantly shorter measurement times and (ii) smaller testing specimens. The smaller testing specimens mean not only a lower consumption of material (which may be an issue for some special systems prepared in a laboratory scale), but also a higher reliability of the results due to the better statistics (with a single specimen for macrocreep it is possible to perform tens or hundreds of microcreep measurements), and a possibility to investigate local changes of the creep behavior (for example in the samples exhibiting core-shell morphology due to aging or oxidation). Therefore, a question arises if it is possible to replace the time-consuming macrocreep tensile experiments with the shorter microcreep indentation experiments, or at least to employ the microcreep measurements in preliminary studies and predictions of macrocreep behavior. According to the available literature, the number of studies comparing the macroscale tensile creep with micro- or nanoindentation creep is quite modest [8,9,10,11,12,13], some of them employs both tensile and indentation creep in macroscale [14,15,16], and just a few articles deals with polymer materials [17,18].

This study is focused on direct comparison of macroscale tensile creep (macrocreep) and microscale indentation creep (microcreep) measurements on three very common polymers—high-density polyethylene (PE), isotactic polypropylene (PP) and atactic polystyrene (PS)—whose properties range from soft and ductile (PE) to hard and brittle (PS). We wanted to answer the following two questions: (i) Is it possible to make qualitative or even quantitative predictions of a long-term macrocreep behavior from short-term microcreep experiments? (ii) Is the widely used power law model (PL) better in the predicting of long-term creep data (for both tensile and indentation experiments) than the more sophisticated elasto-visco-plastic models (EVP)? Moreover, we aimed at the development of a simple, user-friendly software, which will process both tensile and indentation creep data in an efficient, consistent and transparent way.

## 2. Theoretical Background

This section summarizes the key creep relations that were employed in our study. More creep theory is to be found in suitable textbooks [3,5,19]. We start with the description of creep from standard tensile experiments (Section 2.1), continue with specifics of creep determined from indentation experiments (Section 2.2), and finish with a comparison of tensile and indentation creep measurements and processing (Section 2.3). Moreover, the last Section 2.3 discusses briefly why the indentation creep experiments might be suitable for macroscale tensile creep predictions.

### 2.1. Creep in Tensile Experiments

#### 2.1.1. Principle of Experiments and Basic Types of Tensile Creep Curves

Figure 1 shows a typical tensile creep experiment on a polymer system. The principle of tensile creep experiment (Figure 1a) is very simple: a deadweight (with mass *m*) is hung on a rectangular polymer specimen (with length *L*, width *W* and thickness *T*). The acting force *F* = *mg* (where *g* = gravity of Earth) induces stress *σ* = *F*/*A* (where *A* = *W* × *T* = cross-section area) in the specimen.

We measure the specimen deformation in the form of tensile strain *ε_t_* = *ε* = Δ*L*/*L* (where Δ*L* is length change of a testing specimen) as a function of time (*t*). As the deadweight is applied instantaneously, the *σ*-*t* curves are rectangular functions (Figure 1b) and corresponding *ε*-*t* curves (creep curves) show instantaneous elastic deformation followed by viscous (or visco-plastic) deformation (Figure 1c). The creep behavior is influenced strongly by the ratio of the applied stress (*σ*) and yield stress of the polymer (*Y*) as described elsewhere [20,21,22]. Typically, if *σ*/*Y* ratio is below 0.3–0.5, we observe just primary creep (Figure 1c, light red curve). For *σ*/*Y* between 0.3–0.5 and 0.9–1, the creep curves exhibit primary creep, secondary creep, and tertiary creep stage, followed by rupture of the specimen (Figure 1c, medium red curve). For *σ*/*Y* above 0.9–1, the material deforms quickly and breaks in a short time (Figure 1c, dark red curve).

In most of real-life applications, the polymers are used at loads well below their *Y*, where their deformation is slow and reversible (Figure 1c, light red curve). At higher stresses the material deforms irreversibly or even breaks (Figure 1c, medium and dark red curves), which is rather impractical. In this study, all tensile creep experiments were performed at *σ*/*Y* < 0.5, where all investigated polymers exhibited just the primary creep.

#### 2.1.2. General Relations for Tensile Creep

The deformation of materials depends on their elasto-visco-plastic properties. In ideally elastic specimens, the deformation is instantaneous, independent on time (*t*) and proportional just to the applied stress (*σ*):(1)ε=σ/E=Cσ
where *E* and *C* = 1/*E* is the elastic modulus and compliance in tension, respectively. For polymers, which are elasto-visco-plastic materials, Equation (1) describes just their initial elastic deformation. This initial deformation is followed by a time-dependent creep deformation. In general, the total deformation *ε*(*t*,*σ*) consists of three components: (i) *elastic deformation ε*_0_(*σ*) that is instantaneous and reversible, (ii) *viscous deformation ε_v_*(*t*,*σ*) that is time-dependent and reversible, and (iii) *plastic deformation ε_p_(σ,t)* that is time-dependent and permanent. In the mathematical description of creep experiments, it is practical to combine viscous and plastic deformation into one viscoplastic deformation *ε_vp_*(*t*,*σ*):(2)$$\varepsilon \left(\sigma ,t\right)=\varepsilon_{0}\left(\sigma \right)+\varepsilon_{v}\left(t,\sigma \right)+\varepsilon_{p}\left(t,\sigma \right)=\varepsilon_{0}\left(\sigma \right)+\varepsilon_{vp}\left(t,\sigma \right)$$

For polymers, the viscous deformation is usually *viscoelastic* (the elastic component causes reversibility) and the plastic deformation is usually *viscoplastic* (the viscous component causes time-dependent behavior). This is evident from the elasto-visco-plastic models, which are discussed below in the Section 2.1.4. The reason why we combine the viscosity and plasticity contributions (*ε_v_* and *ε_p_*) into one member (*ε_vp_*) is that the standard creep measurement monitors just the overall deformation as a function of time and, as a result, it is not possible to determine which part of the deformation is reversible and which part is permanent. In order to differentiate the reversible and permanent deformation, we need an additional measurement after releasing the load. Such an experiment is called the *creep recovery* or *elastic recovery* or *reverse creep* [23]. In terms of the tensile compliance (which was defined above as *C* = 1/*E* = *ε*/*σ*) Equation (2) can be re-written as:(3)$$C\left(\sigma ,t\right)=C_{0}\left(\sigma \right)+C_{v}\left(t,\sigma \right)+C_{p}\left(t,\sigma \right)=C_{0}\left(\sigma \right)+C_{vp}\left(t,\sigma \right)$$

If the experimental creep curves (Figure 1c) could be fitted with a suitable analytical function, then the prediction of creep behavior would be greatly facilitated. The creep of polymers usually exhibits non-linear viscoelastic behavior [19,24]. Consequently, the linear viscoelasticity theory yields just approximate creep predictive models, while the non-linear viscoelasticity theory failed to give simple creep models of practical value [19]. As a result, there are two main practical approaches how to describe creep of polymer materials: empirical creep equations (Section 2.1.3) and phenomenological descriptions employing spring-and-dashpot models from linear viscoelasticity theory (Section 2.1.4).

#### 2.1.3. Empirical Equations for Tensile Creep

Many equations describing creep behavior of polymer systems as a function of stress (*σ*) and time (*t*) have been proposed [3,24]. Most of them are purely empirical, while others have some theoretical basis. Despite their wide variety of form, they fall into two main classes, corresponding to the two general equations:(4)ε=F(σ,t)
(5)ε=f(σ)h(t)
where *ε* is the deformation and *f*, *h* and *F* are functions of *σ* and/or *t*. The first group of empirical relations (represented by Equation (4)) is more general, but the second group (Equation (5)) is more attractive, because it separates influence of stress and time into two independent functions. Surprisingly enough [24,25], the creep behavior of many polymer systems at lower stresses (*σ*/*Y* < 0.3–0.5) has been described plausibly by a simple power law model (PL):(6a)$$\varepsilon \left(t,\sigma \right)=A_{1}\cdot \sigma^{m}\cdot t^{n}$$
(6b)$$\varepsilon \left(t\right)=A_{2}\cdot t^{n}$$
where *A*_1_, *A*_2_, *m* and *n* are empirical parameters of the investigated material. The PL model (Equation (6a)) accounts for non-linear viscoelastic behavior, as the creep deformation *ε* is proportional to a non-linear function of stress (*f*(*σ*) ∝ *σ^m^*). The second form of PL model (Equation (6b)) corresponds to the situation when we study creep just as a function of time for given stress (i.e., *σ* = constant and *A*_2_ = *A*_1_ ⋅ *σ^m^*). This approach is quite common in the literature [24] and we use it in this work as well.

Although PL model was quite successful in describing creep of many polymer systems, there are good physical reasons for expecting that the creep equations should contain a constant member. Such a member would correspond to instantaneous elastic deformation (*ε*_0_ in Equation (2)). This fact is reflected in a slightly improved empirical model that is called Nutting’s law (NL) [26,27]:(7a)$$\varepsilon \left(t,\sigma \right)=\varepsilon_{0}\left(\sigma \right)+A_{3}\cdot \sigma^{m}\cdot t^{n}$$
(7b)$$\varepsilon \left(t\right)=\varepsilon_{0}+A_{4}\cdot t^{n}$$
where *ε*_0_, *A*_3_, *A*_4_, *m* and *n* are empirical parameters of the investigated material. The parameter *ε*_0_(*σ*) equals to *σ*/*E* if the load is applied instantaneously (cf. Equation (1)), otherwise it can take more complex forms as described elsewhere [27]. In analogy with PL model, Equation (7a) transforms to Equation (7b) if we study creep just as a function time, while the stress is fixed (*A*_4_ = *A*_3_ ⋅ *σ^m^*). It is worth noting that NL model is a special form of Findley power law model (FL) [4,19]:(8)$$\varepsilon =\varepsilon_{0}\left(\sigma \right)f_{0}\left(\sigma \right)+A_{5}\cdot f\left(\sigma \right)\cdot t^{n},$$
where *f*_0_(*σ*) and *f*(*σ*) are material-specific functions. Further work revealed that *f*_0_(*σ*) and *f*(*σ*) are hyperbolic sines of *σ* and two material-specific constants *σ_e_* and *σ_m_* (*f*_0_(*σ*) = sinh(*σ*/*σ_e_*) and *f*(*σ*) = sinh(*σ*/*σ_m_*). We note that FL transforms to NL if *f*_0_(*σ*) = 1 and *f*(*σ*) = *σ^m^* (compare Equations (8) and (7a)) or if we take *σ* as constant (compare Equations (7a,b) and (8)).

#### 2.1.4. Elasto-Visco-Plastic Models for Tensile Creep

The tensile creep can be described not only by simple empirical equations, as shown in the previous section, but also by more sophisticated phenomenological elasto-visco-plastic (EVP) models. Various types of EVP models are covered in most textbooks of polymer physics [23,28], so we will just revise them briefly. In general, the EVP models combine elastic elements (Hookean springs), viscous elements (Newtonian dashpots), and/or plastic elements (such as sliding frictional elements) [18,23,29]. The EVP models employed in this work (Figure 2), which are suitable for creep description, are based on previous studies of Oyen et al. [30,31,32] and Mencik et al. [6,33].

The mathematical definition of the EVP model shown in Figure 2 is given by Equation (9), which describes the creep deformation as a function of time (the meaning of all parameters is described in Figure 2):(9)$$\varepsilon \left(t\right)=\frac{\sigma }{E_{0}}+\sigma \frac{t}{\eta }+\sigma \overset{N}{\underset{i=1}{\sum }}\frac{1}{E_{i}}\left[1-\exp\left(-\frac{t}{\tau_{i}}\right)\right].$$

The right-hand side of the equation consists of three basic terms: elastic term (spring element in Figure 2, representing instantaneous elastic deformation), viscous term (dashpot element in Figure 2, representing viscous deformation that increases with time linearly), and sum of viscoelastic terms (sum of KV elements in Figure 2, representing viscoelastic deformation, which increases with time non-linearly up to a certain limit). By the dividing of both sides of Equation (9) with *σ*, considering the definition of time-dependent creep compliance (*C*(*t*) = *ε*(*t*)⋅*σ*) and instantaneous creep compliance (*C*_0_ = 1/*E*_0_), and defining fluidity (*C_v_* = 1/*η*) we obtain:(10)$$C\left(t\right)=C_{0}+C_{v}t+\overset{N}{\underset{i=1}{\sum }}C_{i}\left[1-\exp\left(-\frac{t}{\tau_{i}}\right)\right].$$

The advantage of EVP models consists in that we obtain some parameters with physical meaning, namely *C*(*t*) and *C*_0_. *C*(*t*) is the tensile creep compliance (and modulus *E*(*t*) = 1/*C*(*t*)) as s function of time. *C*_0_ represents the initial, elastic, time-independent tensile compliance (and modulus *E* = 1/*C*) of the polymer. *C_v_* is the inverse viscosity of dashpot element (*C_v_* = 1/*η*). *C_i_* and *τ_i_* are compliance magnitude and retardation time of *i*-th KV element, respectively. Nevertheless, as discussed by Mencik et al. [6,33], the EVP parameters associated with D and KV elements (*C_v_*, *C_i_* an *τ_i_*) should be taken simply as constants of the model describing the time-dependent behavior of the material.

### 2.2. Creep in Indentation Experiments

#### 2.2.1. Principle of Indentation Creep Experiments and Shape of Indentation Creep Curves

Figure 3 shows a typical indentation creep experiment on a polymer system. The principle of the indentation creep experiment (Figure 3a) is analogous to that of the tensile creep experiment (Section 2.1.1), but the experiment geometry is rather different: a sharp tip (with a high and known elastic modulus) is forced against the polymer surface (with much lower and unknown elastic modulus). The acting force *F* causes the gradual, time-dependent penetration of the tip into the investigated specimen.

In contrast to tensile creep experiments, the loading cannot be applied instantaneously, because current instrumented indenters have to increase *F* at a finite rate. This results in the characteristic trapezoidal loading during the indentation experiments (Figure 3b), which differs from the typical rectangular loading during tensile creep measurements (cf. Figure 1b). As the loading force increases continuously, the indentation creep curves (Figure 3c) do not show instantaneous elastic deformation (cf. Figure 1c). Moreover, the shape of indentation creep curves is basically the same regardless of the loading force (compare Figure 1c and Figure 3c). When *F* increases, the mean contact pressure under the tip (*P_m_* = *F*/*A*, where *A* is the projected contact area of the tip on the specimen surface) remains the same, and the overall character of the deformation does not change, due to the fact that the relatively small tip is surrounded by a large volume of bulk material. This principle holds for all tip geometries on condition that the mean contact pressures, *P_m_*, are higher than the yield stress of the investigated specimen, *Y*, according to the relation *P_m_* > *cY*, where *c* takes the values from ca 1.5 to 3. More details are to be found in the literature [34]. From the point of view or real applications, it is enough to conclude that the above condition (*P_m_* > *cY*; *c* ranging from 1.5 to 3) is fulfilled automatically for all polymer samples and sharp tips unless the loading forces *F* are not extremely small. 

#### 2.2.2. Basic Relations for Indentation Creep

There is a large difference between the basic description of the tensile creep (which were discussed above in Section 2.1.2) and the basic description the indentation creep (which will be summarized in this section). In principle, it would be possible to adjust the general relations from the field of tensile creep (Equations (1)–(3)) for indentation creep in order to separate elastic, viscous and plastic components of the total deformation, but the typical approach is different. This could be attributed both to historical reasons and different nature of indentation experiments. On the other hand, the empirical equations and EVP models describing indentation creep are analogous to those describing tensile creep, as described in the following two sections (Section 2.2.3 and Section 2.2.4).

In the field of non-instrumented microindentation hardness testing (MH), the measured quantity is the size of the imprint of the tip on the surface of the investigated material. From the imprint size, it is possible to calculate hardness (*H*). For polymers, the great majority of experiments were performed with the Vickers tip [35], and so the most frequently reported quantity is Vickers microhardness (*H*_V_). Special MH measurements can yield also microcreep (*C*_V_) and microplasticity (*P*_V_) as explained elsewhere [7]. The basic relation for microcreep determination was introduced by Balta-Calleja and co-workers [36], who studied the time-dependent part of the plastic deformation of the polymer surface under the indenter and noticed that (*H*) decreases with time according to a simple relation:(11)$$H=H_{0}t^{-n},$$
where *H*_0_ is a coefficient reflecting both sample properties and experimental conditions, and *n* is a constant furnishing the rate of creep of the material. The values of *n* for many polymers were collected in the nice and comprehensive review of Flores et al. [37]. In the classical approach, the constant *n* is obtained by measuring of hardness (*H*) at multiple hold times (*t*) and determining the slope of log(*H*)-log(*t*) lines (Figure 4a). In an alternative approach, more compatible with the MHI data evaluation, the hardness values are converted to *h*^2^ (square of the penetration depth) and *n* is determined from the log(*h*^2^)-log(*t*) lines (Figure 4b). The conversion of Equation (11) is straightforward once we know the geometry of the indenter; in the non-instrumented MH testing, the most common is Vickers indenter (*H* = *H*_V_ = *F*/*A_d_* and A_d_ ≈ 26.4*h*^2^; [34]), and we obtain:(12)$$h^{2}=\frac{F}{26.4H_{0}}t^{n}=A_{6}\cdot t^{n},$$
which is equivalent to the well-established power law model that is known from the tensile creep data interpretation (compare Equations (12) and (6b)). The value of creep constant, *n*, which reflects the creep rate in the log–log scale, can be referred to as microcreep, *C*_V_ (sometimes *C*_V_ is marked a *K* [37] or MC [38,39]).

In the field of instrumented microindentation (MHI), the simple evaluation of creep from *h-t* curves is defined in ISO 14577 standard as:(13)$$C_{IT}=\frac{h_{2}-h_{1}}{h_{1}}\times 100\%,$$
where *h*_1_ and *h*_2_ are the penetration depths of the tip at the beginning and at the end of maximal loading, as illustrated in Figure 4c. Alternatively, it is possible to fit all *h* values (measured during the maximum load) in an analogous way to Equation (12), i.e., using the relation *h*^2^ = *const*⋅*t^n^*). This is shown schematically in Figure 4d,e. It is worth noting that for pointed indenters (such as Vickers and Berkovich), it is more correct to fit *h*^2^-*t* data than *h*-*t* data. The reasons will be explained below in Section 2.2.3 and Section 2.2.4.

#### 2.2.3. Empirical Models for Indentation Creep

The empirical equations for tensile creep (Section 2.1.3; Equations (4)–(8)) can be applied to indentation creep as well, under the following two conditions:The tensile strain (*ε*) is replaced by indenter penetration depth (*h*), taking into account the indenter (i.e., tip) geometry (see Equation (14) below).The tensile stress (*σ*) is replaced with mean contact pressure under the indenter (*P_m_* = *F*/*A*; as defined in Section 2.2.1 and elsewhere [34]).

According to rather comprehensive theory [30,32,40,41,42], which is summarized nicely in the studies of Mencik et al. [6,33], the general equation for tensile creep (Equation (4)) should be transformed to the general equation for indentation creep (Equation (14)) as follows:(14)$$h^{m}=K\cdot F\left(P_{m},t\right).$$

The comparison of Equations (4) and (14) shows that the creep strain (ε) changes to *h^m^*/*K*, where constants *m* and *K* a depend on the tip geometry [6]:For spherical tips: *m* = 3/2 and *K* = 3/(4√*r*), where *r* is the tip radius.For pointed tips (Vickers, Berkovich, conical): *m* = 2 and *K* = π/(2tanα), where α is the semi-angle of the indenter tip or equivalent cone (for the most common Vickers and Berkovich tips α = 70.3°).

It is, of course, possible to fit *h*-*t* curves with an arbitrary empirical relation without the recalculations above (and many authors ignore them), but at the cost of lower compatibility between the results from tensile and indentation creep experiments, or between the results from different indentation methods.

Most of the studies dealing with indentation creep of polymers try to predict *h*(*t*) for a given load [43,44,45,46]. The most widely used indenters are Vickers (for microindentation) and Berkovich (for nanoindentation). If we consider the simpler version of the power law from tensile creep theory (Equation (5)) and perform all abovementioned substitutions, we obtain:(15)$$h^{2}=\frac{\pi }{2\tan\alpha }{\left(P_{m}\right)}^{m}t^{n}=A_{7}\cdot t^{n},$$
which is similar to Equation (12) from the non-instrumented indentation. The similarity of Equations (12) and (15) is logical, because both MH and MHI creep experiments measure basically the same quantity.

#### 2.2.4. Elasto-Visco-Plastic Models for Indentation Creep

The elasto-visco-plastic models from tensile creep (Section 2.1.4) can be applied to indentation creep data as well, but the following modifications should be made to obtain physically meaningful parameters, comparable to those from tensile creep:The tensile strength (*ε*) is replaced by penetration depth (*h*) considering the indenter geometry, as described in the previous section (i.e., by means of Equation (14) with correct constants for given indenter).The difference between the instantaneous rectangular loading in tensile creep experiments (Figure 1b) and the finite-rate trapezoidal loading in indentation creep experiments (Figure 3b) must be corrected for, as specified below.

After all theoretical considerations, which have been described elsewhere [6,33], Equation (9) changes as follows in order to describe indentation creep:(16)$$h{\left(t\right)}^{m}=FK\left[C_{0}+C_{v}\left(t-t_{R}/2\right)+\overset{N}{\underset{i=1}{\sum }}C_{i}\left\{1-\rho_{i}\;\exp\left(-\frac{t}{\tau_{i}}\right)\right\}\right],$$
where the constants *m* and *K* come from Equation (14), *F* is the loading force, *t_R_* is the ramping time (i.e., time needed to achieve full loading force, *F*), *ρ_i_* are ramp correction factors defined below (Equation (17)), and the remaining model parameters (*C*_0_, *C_v_*, *C_i_* and *τ_i_*) are compliances and retardation times analogous to those in Equations (9) and (10). The term *t_R_*/2 expresses the fact that the indenter penetration during the load increase period *t_R_* corresponds to the average force *F*/2 [33]. The ramp correction factors, *ρ_i_*, which were introduced by Oyen [30], reflect the difference between rectangular and trapezoidal loading (i.e., they are corrections considering the fact that the time *t_R_* is needed to achieve the full load in indentation creep experiments):(17)$$\rho_{i}=\frac{\tau_{i}}{t_{R}}\exp\left(\frac{t_{R}}{\tau_{i}}-1\right).$$
The constants *C*_0_, *C_v_*, *C_i_*, *τ_i_* and *ρ_i_*, might be obtained by minimizing the sum of the squared differences between the measured and calculated *h*(*t*)*^m^* values in Equation (16), but the actual procedure must be modified. At first, there are several constant terms in Equation (16) that do not depend on time (*C*_0_, *C_v_*, *t_R_*/2, and *C_i_*). Regression analysis cannot determine them individually, but only as a whole. At second, the unknown ramping correction factors, *ρ_i_*, depend on the retardation times *τ_i_*, which are also unknown. Therefore, the Equation (16) must be re-written so that it was suitable for curve-fitting:(18)$$h{\left(t\right)}^{m}=FK\left[B+C_{v}t-\overset{N}{\underset{i=1}{\sum }}D_{i}\exp\left(-\frac{t}{\tau_{i}}\right)\right],$$
where:(19)$$B=C_{0}-C_{v}t_{R}/2+\sum_{i}C_{i}$$
(20)$$D_{i}=C_{i}\rho_{i}$$
The determination of material parameters then proceeds in four steps:Step 1: The calculation of regression constants (*B*, *C_v_*, *D_i_*, and *τ_i_*) by fitting *h*(*t*)*^m^* data with the regression function described by Equation (18).Step 2: The calculation of the ramp correction factors *ρ_j_* from Equation (17) using the known value of *t_R_* (ramping time) and the retardation times *τ_j_* from step 1.Step 3: The calculation of compliances *C_i_* from Equation (20), employing the ramping correction factors from step 2.Step 4: The calculation of compliance *C*_0_ from Equation (19), using the relevant parameters from all previous steps.

The material parameters from Equations (16)–(20) are analogous to the tensile creep parameters from Equations (9) and (10). Moreover, if we evaluate creep data according to Equations (1)–(20), the material parameters and creep constants obtained from tensile creep experiments (Equations (1)–(10)) are comparable to those obtained from the indentation creep experiments (Equations (11)–(20)) as far as their values, units, and physical meaning are concerned. In order to facilitate the tensile and indentation creep comparison, we developed program package MCREEP, which takes all abovementioned relations and equations into consideration. The MCREEP package fits both tensile creep and indentation creep data with power law and elasto-visco-plastic models (more details in Section 3.4).

### 2.3. Comparison of Tensile and Indentation Creep Experiments

Both tensile and indentation creep measurements test, in principle, the same material feature—the resistance to a long-term loading. Nevertheless, the geometry and setup of tensile and indentation creep experiments are different. Moreover, there are some differences due to various historical developments of the two methods. These result in several similarities and dissimilarities of the two methods that are briefly summarized below:*The homogeneity of the stress field* in the two methods differs as a result of different experiment geometry (as evident from the comparison of Figure 1a and Figure 3a). In the tensile creep experiments, the stress within the whole specimen in the same (Figure 5a), while indentation creep experiments exhibit inhomogeneous stress distribution, with a maximum stress just below the tip and gradual non-linear decrease at all directions (Figure 5b). For the sake of completeness, we should note that the stress field during the tensile creep experiments can somewhat change as the specimen is deformed, but the basic difference between homogeneous and non-homogeneous nature of the stress distribution during tensile and indentation experiments is evident and well-documented in the literature [10,47].*The loading rates* in tensile and indentation creep experiments differ due to different experimental setups. In tensile creep measurements, we simply release a deadweight and, as a result, we have (theoretically) instantaneous loading and rectangular loading curve (Figure 1b). In instrumented indentation testers, the load must be increased gradually (due to technical reasons and construction of the devices) and, consequently, we have to use finite loading rates, which results in trapezoidal loading curve (Figure 3b). The deformation in during the loading step of indentation experiments must be corrected for, as described above in Section 2.2.*The applied stress* in indentation creep experiments is usually higher than in tensile creep experiments. The tensile creep measurements are mostly performed at stress (*σ*) well below the material yield stress (*Y*). The reason is shown in Figure 1c—for medium stresses (*σ* > 0.3–0.5*Y*) applied for longer times, the polymer systems tend to deform irreversibly (which is impractical for most applications) and for higher stresses (*σ* > 0.9–1.0*Y*) the deformation is fast and followed by rupture (which means that the tested material cannot be used at given conditions). The indentation measurements (including indentation creep measurements) are usually carried out in the plastic region [34]. In the plastic region, the mean contact pressure (*P*_m_ = *F*/*A*, where *F* = the applied loading force and *A* = the projected area) is proportional to the yield stress of the material (*P*_m_ = *cY*, where *c* = constraint factor, whose values range from 2 to 3 depending on the material and measurement conditions [35]). In other words, the value of mean contact pressure *P*_m_ during the indentation creep experiments is much higher than *Y*, and all indentation creep curves look similar regardless of the loading force *F* (compare Figure 1c and Figure 3c). This is connected with the inhomogeneous stress field in the indentation experiments (Figure 5b) and with the fact that the mean contact pressure in the plastic region does not increase with loading force (*P*_m_ = *cY*, as mentioned above and discussed elsewhere [34,35]).*The evaluation of tensile and indentation creep experiments* is slightly different due to the all abovementioned differences and the separate historical development of tensile and indentation methods: (i) The drastically simplified description of creep by a single number has been established only in the field of the indentation experiments (Section 2.2.2). (ii) The application of empirical models to tensile and indentation creep data is more-or-less the same (Section 2.1.3 and Section 2.2.3). (iii) The application of phenomenological models to tensile and indentation creep data is analogous, but we have to use correction factors for the finite loading rates and different geometry during the indentation creep measurements (Section 2.1.4 and Section 2.2.4).*The correlations among the indentation and tensile creep measurements* for polymer systems is an open question and the subject of this study. On one hand, we might suppose that various types of creep should be proportional to each other. On the other hand, all abovementioned differences, together with the fact that polymers exhibit complex elasto-visco-plastic behavior, indicate that the relations between tensile and indentation creep may not be so straightforward.

## 3. Experimental

### 3.1. Materials

The macroscale uniaxial tensile creep tests and microscale indentation creep tests were carried out on three different polymers: PE, (high-density polyethylene, Liten MB62, by Unipetrol RPA; Zaluzi, Czech Republic) characterized by melt flow index MFI (190 °C, 5 kg) = 20 g/10 min (ISO 1133); density (ISO 1183) 960 kg/m^3^; weight-average molar mass M_w_ 420,000 g/mol; yield stress of 24 MPa, PP (isotactic polypropylene, Mosten TB003, by Unipetrol RPA): MFI (230 °C, 2.16 kg) = 3.2 g/10 min; density 946 kg/m^3^; M_w_ 420,000 g/mol; yield stress = 34 MPa and PS (atactic polystyrene, Krasten 171, by SYNTHOS Kralupy a.s.; Kralupy n. Vlt., Czech Republic): MFI (200 °C, 5 kg) = 1.7 g/10 min; density 1050 kg/m^3^; M_w_ 400,000 g/mol; and yield stress = 50 MPa. 

### 3.2. Sample Preparation

The macrocreep testing specimens were prepared by injection molding according to the standard ČSN EN ISO 5247-2-1A. The granulates of PE, PP, and PS were molten and injected into the standard mold attached to the injection molding machine (ENGELSPEX Victory 50; Engel, Austria). The temperature profiles used for the injection molding process of the selected polymers are given in Table 1.

### 3.3. Creep Measurements

#### 3.3.1. Macroscale Tensile Creep

Uniaxial tensile creep was measured using a laboratory-made device with a deadweight fixed on a mechanical stress amplifier (lever) 10:1. A digital strain gauge (with accuracy of about 1 µm) was connected with the upper clamp of the specimen to record the displacement. The samples were prepared from injected dumb-bell specimens with initial distance between grips 100 mm and cross-section 4 mm × 10 mm. All specimens for creep measurement were stored in dark at room with the stable temperature (~22 °C). Mechanical pre-conditioning before each creep measurement consisted in one-minute application of stress, which produced a strain larger than the expected final strain attained in the following creep experiment; the recovery period after the pre-conditioning was 1 h. Series of creep measurements with the hold time of 100 min were performed at four gradually increasing stress levels with one test specimen. The applied stresses were below the yield stress of each material to avoid plastic deformation or breaking of the specimen. The selected loads are listed in Table 2. Each measurement was followed by a 22 h recovery before another creep test (at an increased stress) was initiated. The creep measurements were performed in an air-conditioned room (nominal temperature 22 °C) and the creep devices were placed in a closed wooden box padded with 2 cm of foamed polystyrene at all sides (including the opening doors).

The macroscale tensile creep data were processed in two ways. Firstly, the *ε*-*t* curves (elongation as a function of time) were employed in calculation of creep constant, analogous to the creep constant from MHI experiment. Secondly, PL and EVP models were fitted to the *ε*-*t* curves. 

#### 3.3.2. Microscale Indentation Creep

Microscale indentation creep measurements were carried out with both non-instrumented microindentation hardness tester (VMHT Auto Man; UHL, Germany) and instrumented microindentation hardness tester (MCT tester; CSM, Switzerland). Both non-instrumented and instrumented microindentation hardness testing experiments were carried out using the Vickers method: a diamond square pyramid (with the angle between non-adjacent faces 136°) was forced against the flat surface of a specimen. Smooth surfaces for the microindentation experiments were prepared by a microtomy: the macroscale dumb-bell specimens (described in Section 3.2) were cut perpendicularly with a rotary microtome (RM 2155; Leica; Vienna, Austria), using a commercial microtome blade in the case of softer samples (PE and PP) or a laboratory-made freshly-broken glass knife (Leica EM KMR3; Leica, Austria) in the case of the hardest sample (PS). For both methods (MH and MHI) and all three samples (PE, PP and PS), the experiments were performed at two different loading forces: *F* = 50 gf (490 mN) and 200 gf (1961 mN). For each specimen (i.e., each smooth surface), at least 10 indentations were carried out and the results were averaged.

In the case of non-instrumented microindentation (MH), we used two loading forces specified in the previous paragraph (50 and 200 gf) and instant loading and unloading. The values of Vickers microhardness (*H*_V_) and corresponding penetration depths (*h*) of the indents were calculated from the measured diagonals lengths of the indent [38,48]. Microcreep (or creep constant) was determined from *h* values that were measured at five different dwell times (6, 12, 24, 48 and 96 s) as described above (Section 2.2.2, Figure 4).

In the case of instrumented microindentation (MHI), we used the same loading forces as in MH measurements (50 and 200 gf) and maximum linear loading rate, which did not cause loading artifacts, such as short overloading. This maximal loading rate in our instrumented device corresponded to total loading/unloading time of 2 s, which yielded linear loading rates of 14,715 and 58,860 mN/min for 50 and 200 gf, respectively. The MHI results were processed in three independent ways as far as creep characterization is concerned. At first, we calculated the simple values of indentation creep, *C*_IT_, from the penetration depths at the beginning and at the end of the indentation experiment, respectively (according to simple formula defined in ISO 14577; Section 2.2.2; Equation (13)). At second, we determined creep constant from *h*-*t* curves (penetration depth, *h*, as a function of time, *t*), where the creep constant was coefficient *n* from power law (as explained in Section 2.2.2, Equation (12) and Section 2.2.3, Equation (15)). At third, PL and EVP models were fitted to *h*-*t* curves (as described in detail in Section 2.2.3, Equation (15) and Section 2.2.4, Equations (16)–(20)). For each sample (PE, PP and PS) we measured and evaluated both short-term creep (total time of maximal load 100 s) and long-term creep (total time of maximal load of 3000 s = 50 min).

### 3.4. Creep Data Evaluation

For fast and convenient fitting of creep data with various models, we developed the program package MCREEP. The package works with both tensile and indentation creep data. It fits the data with both PL and all EVP models that are employed in this work. The MCREEP is a freeware Python package, whose source code, documentation and installation instructions are available in the standard Python www-repository (Python Package Index; https://pypi.org/project/mcreep (accessed on 10 January 2023)).

## 4. Results

### 4.1. Short-Term Indentation Creep

Figure 6 shows the fitting of creep predictive models to experimental *h*-*t* curves from short-term indentation creep measurements (hold times 100 s) of all investigated polymers (PE, PP, and PS). The tested predictive models comprised three phenomenological EVP models (S + D + 1 KV, S + D + 2 KV and S + D + 3 KV, illustrated in Figure 2 and described by Equation (16) with *N* = 1, 2 and 3) and one empirical model (power law; Equation (15)). The short-term indentation creep was carried out for 100 s at two loading forces, *F* = 50 gf and *F* = 200 gf, but Figure 6 shows just the data for 50 gf for the sake of brevity (the results for 200 gf were analogous as documented in Appendix B). Each short-term indentation creep measurement (i.e., each MHI measurement for given combination of polymer and loading) was performed >10 times. The multiple measurement of one specimen is a common practice in the field of micro- and nanoindentation, because it improves reproducibility and minimizes the risk that the micromechanical properties are determined from a location that is not typical for the whole sample. The averaged results of all measurements for all three samples and both loadings (50 and 200 gf) are summarized in Appendix B, and the complete fitting results of all individual measurements are in Appendix A.

In the case of short-term indentation experiments, the fitting of the predictive models to creep data was performed in the whole time range. Under these conditions, all four models were able to describe the measured data almost perfectly. All EVP models (Figure 2) consisted of one spring element (S-element, corresponding to instantaneous elastic deformation and characterized by parameter *C*_0_), one dashpot element (D-element, corresponding to linearly increasing viscous deformation and characterized by parameter *C_v_*), and finally 1–3 Kelvin–Voigt elements (KV-elements, corresponding to non-linearly increasing viscoelastic deformation and characterized by parameters *C_i_* and *τ_i_* for *i*-th element). This means four parameters for the simplest S + D + 1 KV model, six parameters for S + D + 2 KV model, and eight parameters for S + D + 3 KV model. As the number of parameters increased, the EVP models fitted experimental data more and more accurately, which is documented by the increasing *R*^2^ values in Figure 6 (*R*^2^ are the coefficients of determination [49]). Nevertheless, even the four-parameter S + D + 1 KV model fitted the short-term creep data very well for all three polymers. For the six-parameter and eight-parameter S + D + 2 KV and S + D + 3 KV models, respectively, the fits could even become unstable due to mutual correlations between too many parameters. The stability of the fits could be restored by fixing the retardation times to suitable values covering the investigated time range, as recommended elsewhere [6,33]. The simple, empirical two-parameter power law model fits the short-term creep data equally well, perhaps with the small exception of the very beginning of the *h*-*t* curves, where the higher-parameter EVP models show higher flexibility.

### 4.2. Long-Term Indentation Creep

Figure 7 shows the fitting of creep predictive models to experimental *h*-*t* curves from long-term indentation creep measurements (hold times 3000 s) of all investigated polymers (PE, PP, and PS). The predictive models (three phenomenological EVP models and empirical PL model) and experiment setup were the same as for the short-term creep experiments (Section 4.1), the measurements were performed for two loadings (50 gf and 200 gf), and each sample/loading combination was measured >10 times. Nevertheless, the hold time was longer (100 s vs. 3000 s), and the fitting strategy was slightly different. The averaged results of all long-term indentation creep measurements for all three samples and both loadings are summarized in Appendix C, and the complete fitting results of all individual measurements are in Appendix A.

In the case of long-term indentation experiments, the models were fitted only to the first 600 s of experimental data. This enabled us to compare how well the models can predict long-term creep. All EVP models were able to approximate the first part of *h-t* curves quite well, but overestimated the creep deformation for longer times. The overestimation decreased with the increasing number of KV-elements in the EVP models, i.e., in the following order: S + D + 1 KV < S + D + 2 KV < S + D + 3 KV, but even the best, eight-parameter model S + D + 3 KV, performed poorly as far as the long-term creep data prediction was concerned. The same trend was observed for all three polymers, starting from the softest and most creeping PE (Figure 7, upper row), through intermediate PP (Figure 7, middle row), to the stiffest and least creeping PS (Figure 7, lower row). In contrast, the simple two-parameter PL model was able to predict long-term behavior quite reliably, as documented also by *R*^2^ coefficients, which were quite close to 1 for all three polymers.

It is worth noting that the coefficients of determination, *R*^2^, which are shown in Figure 6, Figure 7 and Figure 8, were calculated for the whole dataset (i.e., for the data in whole time range). Nevertheless, Appendix B, Appendix C and Appendix D show two values of the coefficients od determination, *R*^2^ (fit) and *R*^2^ (all). The *R*^2^ (fit) coefficients were calculated just for the initial time range, within which the fitting was performed (i.e., for the initial 600 s of experimental data, to which the models were fitted in the case of long-term creep measurements), while the *R*^2^ (all) coefficients were calculated for the whole data range (like *R*^2^ in Figure 6, Figure 7 and Figure 8).

### 4.3. Long-Term Tensile Creep

Figure 8 shows the fitting of creep predictive models to experimental *ε*-*t* curves from long-term tensile creep measurements (hold times 6000 s) of all three investigated polymers. The predictive models were the same as for the indentation creep (Section 4.1. and Section 4.2), but they included all adjustments and recalculations described above (Section 2) in order to achieve maximum compatibility between tensile and indentation creep results. For each polymer, we used four different loadings (Table 2), which were gradually increasing, but they were kept sufficiently below the yield stress, *Y*, of the individual polymers, in order to avoid rupture of the specimen (Section 3.3.1). Each measurement (i.e., each combination of polymer/loading) was performed at least two times (additional measurements were added if the first two measurements were too different) and the results were averaged. Figure 8 shows just the fitting for the highest loads applied to each polymer (PE: *Y* ≈ 20 MPa, load = *σ* = 9.81 MPa; PP: *Y* ≈ 30 MPa, load = *σ* = 14.72 MPa; PS: stress at break ≈ 40 MPa, load = *σ* = 14.72 MPa). The results for the lower loads were analogous. The averaged results of all long-term tensile creep measurements for all three samples are summarized in Appendix D, and the complete fitting results of all individual measurements for all materials and loadings are in Appendix A.

In the case of long-term tensile creep experiments, the models were fitted only to the first 600 s of experimental data (as in the case of long-term indentation creep experiments, which are described in the previous section). PS showed very low creep in tension, which resulted in a small, noisy, and stepwise increase in tensile strain (Figure 8, lower row, values of *ε_t_*), but the experimental data were still good enough for fitting (Figure 8, lower row, dotted regression curves). In general, the fitting of creep models to long-term tensile creep data and long-term indentation creep showed analogous trends (cf. Figure 7 and Figure 8). The EVP models failed to predict the long-term creep behavior, albeit the predictions tended to improve with the increasing number of KV-elements in the row: S + D + 1 KV < S + D + 2 KV < S + D + 3 KV. The simple PL model predicted the long-term creep data quite successfully, as documented by the visual comparison of the experiment and regression curves, and the much lower values of *R*^2^ in comparison with EVP models (Figure 8, the rightmost column).

### 4.4. Additional Indentation Creep Results

In addition to the instrumented microindentation (Section 4.1 and Section 4.2) and instrumented tensile testing (Section 4.3), we performed two additional creep measurements and/or evaluations:From MH (non-instrumented microindentation) experiments, we evaluated creep constants as described in Section 2.2.2. The principle of the evaluation is shown in Figure 2a,b, above, and the final creep constant values are given in Table 3, below, together with the creep constants from all instrumented creep measurements.From MHI (instrumented microindentation) experiments, we calculated the values of indentation creep, *C*_IT_, according to ISO 14577. The principle of the evaluation is shown in Figure 2c, above, and the final *C*_IT_ values are given in Table 4, below.

### 4.5. Quantification of Creep Behavior by Means of Creep Constants

The creep of all three investigated polymers (PE, PP, and PS) could be characterized by ten independent values, as we used three independent methods (non-instrumented indentation, instrumented indentation, and tensile creep measurements), two times for instrumented indentation (shorter- and longer-time indentation creep) and several loads (two loads for indentation measurements and four loads for tensile creep measurements). Each quantification yielded a specific creep constant, which was proportional to the creep rate (i.e., to the increase in the specimen deformation with time). In fact, the creep constants are the creep exponents *n* that were obtained from fitting the power law to experimental creep data (Equation (6b) for tensile creep, Equation (12) for MH creep data, and Equation (15) for MHI creep data). The creep constants from all experiments are collected in Table 3.

Moreover, the instrumented indentation measurements yield the value of indentation creep, *C*_IT_. Unlike the creep constants that are proportional to the creep rate, the *C*_IT_ values are proportional to the total creep deformation (as illustrated in Figure 4c). The *C*_IT_ values are summarized in Table 4. 

Although the characterization of creep behavior by a single value may look like a rough approximation, it is well-established in the literature. The *C*_IT_ evaluation is incorporated in the ISO 14577 standard, while the creep constants (i.e., the creep exponents, *n*) have been introduced in the pioneering studies of Balta-Calleja and co-workers [35,37]. For each combination of measurement and evaluation method, the creep constants in Table 3 and Table 4 show the same trend and decrease in the following order: PE > PP > PS. In other words, all experiments were in agreement that PE exhibited the highest creep, PP showed medium creep, and PS displayed the lowest creep values. This is one of the key results of our study, which is discussed in detail in the following section. 

## 5. Discussion

### 5.1. Correlations between Indentation Creep and Tensile Creep Results

The first objective of this work was to verify if the short-term microscale indentation creep experiments can be employed in predicting long-term macroscale tensile creep properties of polymer materials. According to available literature, which is summarized in the Introduction (Section 1), there are very few studies on this topic. Lu et al. [17] studied creep compliance of two amorphous, stiff, brittle, glassy polymers (poly(methyl methacrylate), PMMA, and poly(carbonate), PC). The authors concluded that the results from their nanoindentation tests generally agreed with the results from conventional tension and shear tests on PMMA and PC, which were performed in previous studies. Peng et al. [18] studied non-linear viscoelastic behavior of polypropylene (PP) by means of both macroscale uniaxial creep and nanoindentation creep measurements. Both methods confirmed the non-linear viscoelasticity. The authors observed and discussed non-uniform local properties of the PP surface and interpreted them as an intrinsic property of the samples, without considering possible indentation size effects. Other studies dealing with both macro- and microcreep were performed on inorganic materials. In this contribution, we compared macroscale tensile creep (macrocreep) and microscale indentation creep (microcreep) on three common polymers with very different properties: from the softest and most ductile PE, through intermediate PP, to stiff and brittle PS. We employed multiple creep models and the state-of-the art theory in order to characterize the tensile and indentation creep in the mutually compatible way (Section 2, Equations (1)–(20)). On the condition that all conversions and corrections (such as tip geometry or ramp correction factors) are considered, the tensile and indentation creep parameters showed not only excellent qualitative, but also fair quantitative agreement, as documented in Figure 9 and Figure 10.

Figure 9 summarizes the results of the creep rate evaluation by means of the empirical power law model fitted to both microcreep and microcreep data (Figure 9a) and a straightforward calculation of indentation creep, *C*_IT_, according to ISO 14577 standard (Figure 9b). The results of all three methods (MH, MHI, and tensile creep measurements) and evaluations (creep constants, *n*, and indentation creep values, *C_IT_*) are in unambiguous agreement that the creep decreases in the order PE > PP > PS. Moreover, there is also fair agreement between the values of creep constants, at least for PE and PP. The stiff and brittle PS polymer exhibited much lower tensile creep in comparison with PE and PP, while the difference in the indentation creep constants among the three polymers was not so huge. This high difference between tensile and indentation creep rate for the stiff polymers is a subject of our ongoing research. Preliminary results suggest that the phenomenon is connected with the inhomogeneity of the stress field underneath the indenter during the indentation experiments (Figure 5).

Figure 10 displays the values of instantaneous creep compliance, *C*_0_, which was estimated from the fitting of our most advanced EVP model (Figure 2c, model S + D + 3 KV) to MHI and tensile creep data. Numerical values of all regression parameters are summarized in Appendix B, Appendix C and Appendix D. Nevertheless, as discussed elsewhere [6,33], only *C*_0_ parameter has direct physical meaning, being inversely proportional to the instantaneous elastic modulus of the material (*E*_0_ = 1/*C*_0_). The remaining parameters (*C_v_*, *C_i_* and *τ_i_*) may characterize some specific processes in the material, but, in general, they should be regarded simply as constants in a mathematical model able to describe the material response in some time interval with some accuracy. In any case, the values of *C*_0_, obtained from fitting of S + D + 3 KV model to short-term indentation creep data, long-term indentation creep data, and tensile creep data (Figure 10), exhibited the perfect qualitative trend (*C*_0_(PE) > *C*_0_(PP) > *C*_0_(PS)) for all eight independent measurements, reasonably similar values for all eight measurements and all three polymers, and also very good agreement with tabulated elastic moduli of the three polymers (*E*_0_(PE) ≈ 1 GPa ⇒ *C*_0_(PE) ≈ 1 GPa; *E*_0_(PP) ≈ 1.3 GPa ⇒ *C*_0_(PP) ≈ 0.8 GPa; *E*_0_(PS) ≈ 3 GPa ⇒ *C*_0_(PS) ≈ 0.3 GPa). These encouraging results indicated that the shorter-term indentation creep experiments could be employed in predicting longer-term tensile creep data of polymer systems.

### 5.2. Comparison of Predictive Models

The second objective of this work was to compare the creep predictions from simple, widely-used, empirical PL model and more sophisticated, multiparameter, phenomenological EVP models. The extensive testing (Figure 6, Figure 7 and Figure 8; Appendix B, Appendix C and Appendix D) of all abovementioned models could be summarized as follows:For the short-term creep description, all four investigated models, i.e., the empirical power law model and all three elasto-visco-plastic models, performed very well.For the long-term creep predictions, the empirical PL model performed better than more sophisticated phenomenological EVP models. This is clearly documented in Section 4.2 and Section 4.3. The EVP models contained higher number of parameters (from four to eight), but the simple, two-parameter PL model exhibited better results. This explains why the PL model is so popular in both tensile and indentation creep predictions.

### 5.3. Automated Creep Data Evaluation with Package MCREEP

The Python package MCREEP is an important additional result of this study. It brings consistent, reproducible, and standardized processing of both tensile and indentation creep data. The tensile creep data are evaluated according to ISO 899, while the indentation creep data are processed according to the theoretical background and standardized protocols established in the publications of Oyen et al. [30,32] and Mencik et al. [6,33]. The MCREEP calculations consider all relations and corrections, which enabled us to compare tensile and indentation data not only qualitatively (Figure 6, Figure 7 and Figure 8), but also quantitatively (Figure 9 and Figure 10). In fact, all calculations, tables and figures in the Results section of this contribution were made with the MCREEP package. The package is a free software, focused on fast, easy, and user-friendly processing of experimental data. More details and installation instructions are available in Section 3.4.

### 5.4. Limitations of Current Study

Our work has demonstrated that the results of short-term microscale indentation creep experiments can be employed in the predictions of the long-term macroscale tensile creep behavior of polymers, at least on qualitative level. According the available literature, this is the very first systematic comparison of macroscale tensile creep and microscale indentation creep of polymer materials. Nonetheless, the study has several limitations and/or issues, namely:The long-term macroscale creep was measured with holding time 6000 s = 100 min. This was more than sufficient for the comparison of micro- and macroscale creep, but tensile creep experiments are usually performed at higher times. Long-term macroscale creep experiments are ongoing in our laboratory. The preliminary results seem to be in agreement with current findings, but the experiments have not been finished yet.Although the three investigated polymers covered broad range of properties—PE was very soft and ductile, PP was stiffer and harder, and PS was very stiff, hard and brittle—more polymers should be tested to verify our results. Moreover, the next research in this field should cover also polymer blends and composites.For creep predictions, we focused our attention to the most popular models. From the empirical models, we selected the power law model (Section 2.1.3 and Section 2.2.3), and from EVP models, we employed those suitable and recommended for indentation creep experiments (Section 2.1.4 and Section 2.2.4). The tested higher-parameter EVP models (with two and three KV elements) exhibited somewhat lower stability during the fitting (a necessity to fix retardation times or to provide their estimates so that the fitting converged). The limited stability of the higher-parameter models is documented also in the Appendix A, where we calculated the covariance matrices for selected representative cases, showing that the covariances among parameters increased in the row S + D + 1 KV < S + D + 2 KV < S + D + 3 KV. Nevertheless, it is possible that some other higher-parameter models, such as generalized fractional models [50,51], might perform better.

## 6. Conclusions

We compared macroscale tensile creep (macrocreep) and microscale indentation creep (microcreep) for three common polymers: PE, PP and PS. The creep was characterized by three independent methods: non-instrumented microindentation, instrumented microindentation, and instrumented tensile creep measurements. Various models were used to fit both tensile and indentation creep data. The most important results can be summarized as follows:All macrocreep and microcreep measurements were in agreement that the creep of the three investigated polymers decreased in the order: PE > PP > PS. This suggested that short-term microindentation creep experiments might be used for prediction of long-term macroscopic creep behavior of polymer systems.The fitting of various predictive models to both macrocreep and microcreep data documented that the simple empirical two-parameter power law model (creep deformation ≈ *A*⋅*t^n^*) yielded better predictions of long-term creep behavior than the more sophisticated, multiparameter elasto-visco-plastic models.We developed a freeware program package MCREEP for fitting the abovementioned theoretical models to experimental creep data, considering the key differences and relations between tensile and indentation creep. It has been demonstrated that fitting of tensile and indentation creep data with MCREEP yields *quantitatively* comparable results, such as similar values of creep constants and elastic moduli.

## Figures and Tables

**Figure 1 materials-16-00834-f001:**
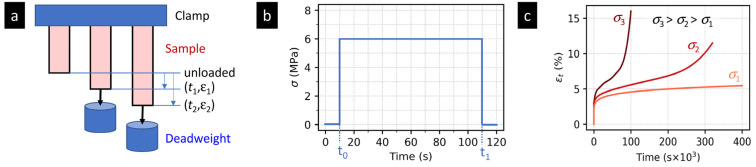
Scheme of a typical tensile creep experiment with a polymer sample: (**a**) the experiment geometry, (**b**) the typical sample loading as a function of time = the *σ*-*t* curve, (**c**) the typical creep curves (*ε*-*t* curves) showing the time dependence of strain, ε, for three different levels of loading. The creep curves were redrawn according to our measurements and available literature [20,21,22].

**Figure 2 materials-16-00834-f002:**
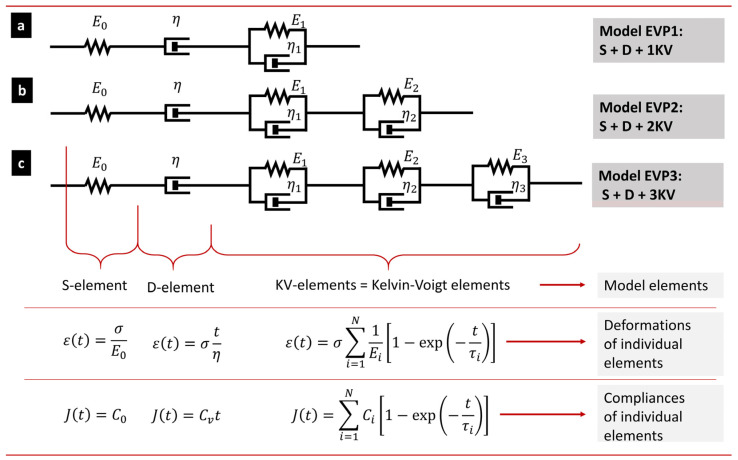
EVP models of a polymer system, which are suitable for the creep description. The model consists of elastic elements (springs; S), viscous elements (dashpots; D) and several viscoelastic elements (spring and dashpot in parallel configuration; Kelvin–Voigt element; KV). The number of KV elements can be adjusted to given experiment; for common creep experiments it usually ranges from one (**a**) to two (**b**) or three (**c**) [6]. We note that possible plastic deformation during the tensile creep experiment is comprised in the D-element (which represents, in this case, also irreversible deformation).

**Figure 3 materials-16-00834-f003:**
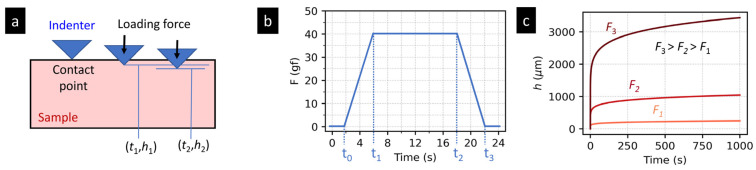
Scheme of a typical indentation creep experiment with a polymer sample: (**a**) the experiment geometry, (**b**) the typical sample loading as a function of time = the *F*-*t* curve, (**c**) the typical creep curves (*t*-*h* curves) showing the time dependence of the tip penetration depth, *h*, for three different levels of loading. The curves were taken from our parallel measurements on PE sample (unpublished results).

**Figure 4 materials-16-00834-f004:**
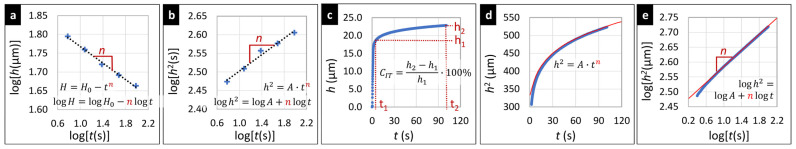
Basic relations for creep in (**a**,**b**) non-instrumented MH testing and (**c**,**e**) instrumented MHI testing. The experimental data in all graphs (blue points and curves) come from real indentation experiments on PE. In MH testing, the value of creep constant, *n*, can be determined from measurement of hardness, *H*, for several hold times, *t*. The value of creep constant can be determined either from (**a**) log(*H*)-log(*t*) relation according to Equation (11), or (**b**) from log(*h*^2^)-log(*t*) relation according to Equation (12). In MHI testing, the creep can be characterized either (**c**) by a simple evaluation of penetration depths according to Equation (13) or (**d**) from the fitting power law model on the whole *h*-*t* curve according to Equation (15); the fitting of power law to MHI data (**e**) can be performed in log(*h*^2^)-log(*t*) scale as well, in analogy with MH—compare subplots (**b**) and (**e**).

**Figure 5 materials-16-00834-f005:**
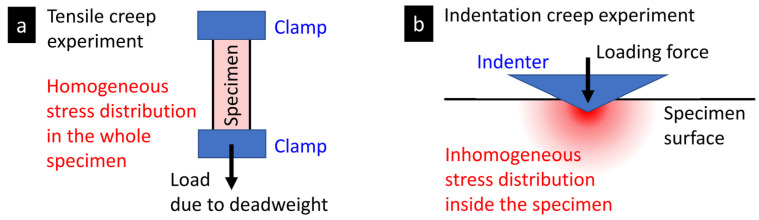
The difference between (**a**) tensile creep and (**b**) indentation creep measurement from the point of view of experiment geometry and homogeneity of the stress field in the specimen.

**Figure 6 materials-16-00834-f006:**
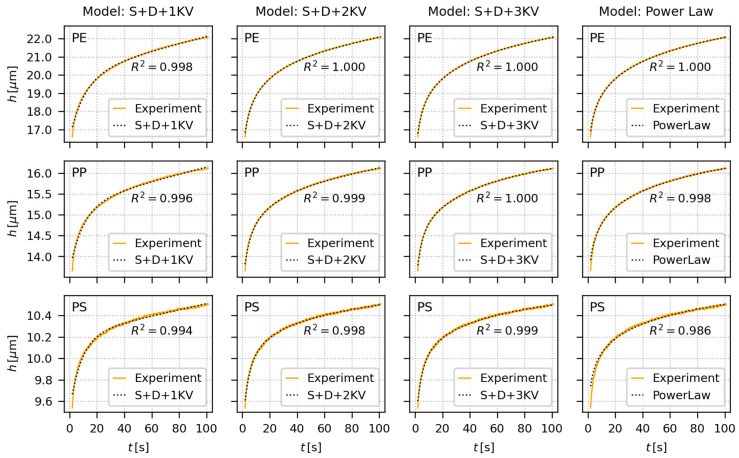
Fitting of creep predictive models (dotted black curves) to representative short-term indentation creep data (orange curves) measured at *F* = 50 gf. The short-term indentation creep measurements were performed for 100 s and the models were fitted to the data in the whole time range. The tested polymers were PE (upper row), PP (middle row), and PS (lower row). The phenomenological, spring-dashpot based EVP models (the first three columns) are illustrated in Figure 2 and described by Equations (16)–(20). The empirical power law model (the last column) is described by Equation (15). Each plot shows coefficient of determination, *R*^2^, which may range from negative values (extremely bad fit), through zero (poor fit) to +1 (perfect fit).

**Figure 7 materials-16-00834-f007:**
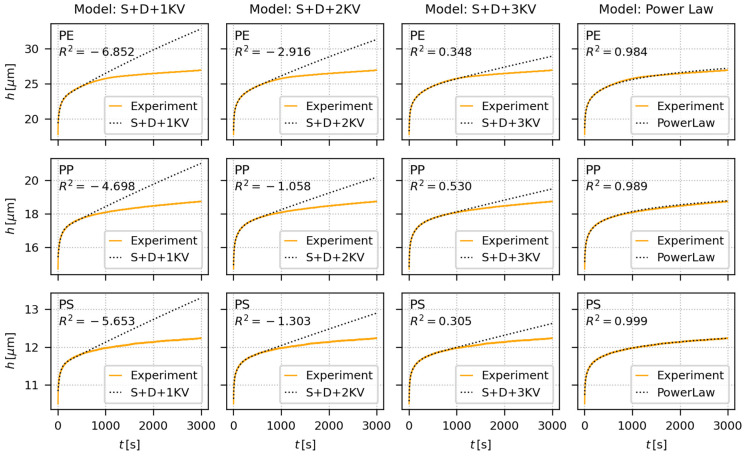
Fitting of creep predictive models (dotted black curves) to representative long-term indentation creep data (orange curves) measured at *F* = 50 gf. The long-term indentation creep measurements were performed for 3000 s and the models were fitted to the data up to 600 s. The tested polymers were PE (upper row), PP (middle row), and PS (lower row). The phenomenological, spring-dashpot based EVP models (the first three columns) are illustrated in Figure 2 and described by Equations (16)–(20). The empirical power law model (the last column) is described by Equation (15). Each plot shows coefficient of determination, *R*^2^, which may range from negative values (extremely bad fit), through zero (poor fit) to +1 (perfect fit).

**Figure 8 materials-16-00834-f008:**
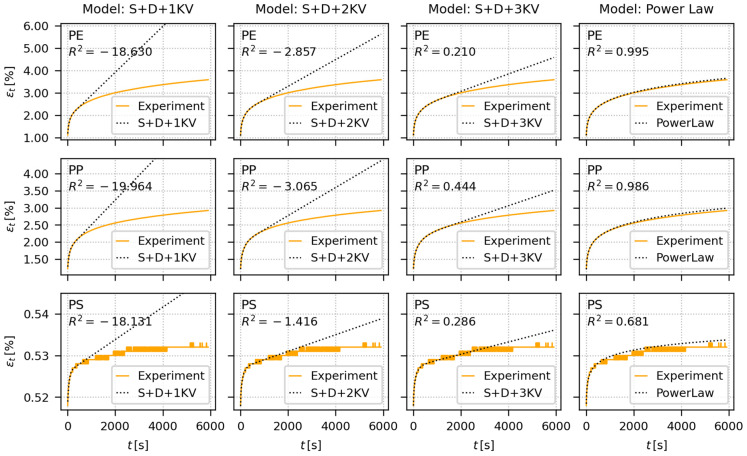
Fitting of creep predictive models (dotted black curves) to representative long-term tensile creep data (orange curves). The long-term tensile creep measurements were performed for 6000 s and the models were fitted to the data up to 600 s. The tested polymers were PE (upper row; load 4 kg ⇒ *σ* = 9.81 MPa), PP (middle row; load = 6 kg ⇒ *σ* = 14.72 MPa), and PS (lower row; load = 6 kg ⇒ *σ* = 14.72 MPa). The phenomenological, spring-dashpot based EVP models (the first three columns) are illustrated in Figure 2 and described by Equations (9) and (10). The empirical power law model (the last column) is described by Equation (6b). Each plot shows coefficient of determination, *R*^2^, which may range from negative values (extremely bad fit), through zero (poor fit) to +1 (perfect fit).

**Figure 9 materials-16-00834-f009:**
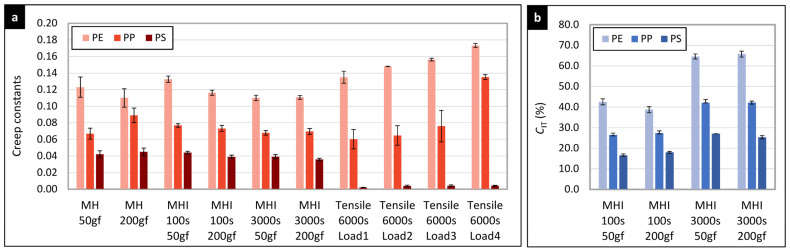
Simple quantification of the indentation and tensile creep of all three investigated polymers: (**a**) the values of creep constants (exponents n from the power law equation; Equations (6b), (12), and (15) and (**b**) the values indentation creep, *C*_IT_, from Equation (13). The error bars represent standard deviations.

**Figure 10 materials-16-00834-f010:**
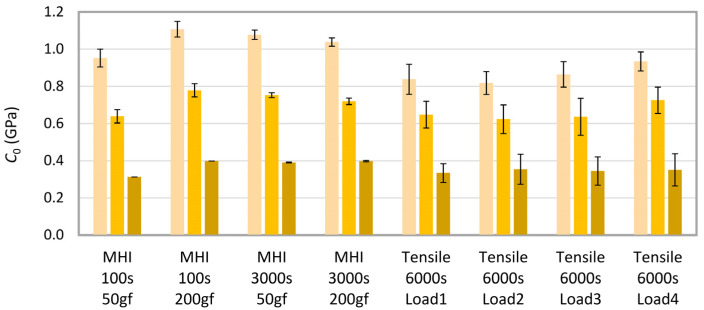
Instantaneous creep compliances, *C*_0_, estimated from fitting of eight-parameter EVP model (S+D+3KV) to indentation and tensile creep data. The error bars are standard deviations.

**Table 1 materials-16-00834-t001:** Temperature profile used for plasticization and preparation of samples for creep testing.

Sample	Zone 3 (°C)	Zone 2 (°C)	Zone 1 (°C)	Die (°C)	Mold (°C)
PE	185	180	180	190	40
PP	240	235	225	240	55
PS	225	220	220	230	35

**Table 2 materials-16-00834-t002:** The loads and corresponding stresses that were used in tensile creep experiments.

Sample	Load 1		Load 2		Load 3		Load 4	
	m (kg)	σ (MPa)	m (kg)	σ (MPa)	m (kg)	σ (MPa)	m (kg)	σ (MPa)
PE	2	4.91	2.5	6.13	3	7.36	4	9.81
PP	2	4.91	3	7.36	4	9.81	6	14.72
PS	2	4.91	4	9.81	5	12.26	6	14.72

**Table 3 materials-16-00834-t003:** Summary of creep constants from all indentation and tensile creep measurements.

Sample	MH		MHI @ 100 s		MHI @ 3000 s		Tensile Creep Test *			
	50 gf	200 gf	50 gf	200 gf	50 gf	200 gf	Load1	Load2	Load3	Load4
PE	0.123	0.110	0.133	0.116	0.110	0.111	0.135	0.148	0.156	0.173
PP	0.067	0.089	0.077	0.073	0.068	0.070	0.060	0.065	0.076	0.135
PS	0.042	0.045	0.044	0.039	0.039	0.036	0.002	0.004	0.004	0.004

* All creep constants were obtained by fitting the creep data with power law; the creep constants are exponents *n* from the Equations (6b) and (12) for tensile and indentation creep, respectively. MH creep tests were performed at two different loads (50 and 200 gf). MHI tests were performed at two different loads (50 and 200 gf) and two different hold times (100 s and 3000 s). Tensile creep tests were performed at four different loads for 6000 s; the loads were increasing in a specific way for each polymer as explained in Section 3.3.1 and the values of all loads for each polymer are summarized in Table 2.

**Table 4 materials-16-00834-t004:** Summary of *C*_IT_ values from all MHI measurements.

Sample	*C*_IT_ (%) from MHI @ 100 s		*C*_IT_ (%) from MHI @ 3000 s	
	50 gf	200 gf	50 gf	200 gf
PE	42.6	38.7	64.6	65.7
PP	26.5	27.4	42.1	42.2
PS	16.6	18.0	27.1	25.3

*C*_IT_ values were determined according to ISO 14557 standard from MHI experiments performed at two different hold times (=times for which the loading force was applied; 100 s and 3000 s).

## Data Availability

The data are available at request to the corresponding author.

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
