# Peer review of "Correlations between Microscale Indentation Creep and Macroscale Tensile Creep of Polymers"

_materials, 2023, doi:10.3390/ma16020834_

Round 1

Reviewer 1 Report

This paper investigates the correlation between indentation creep and the long-term tensile creep behavior among three polymer materials. Empirical models are fit to find the corresponding creep parameters. My major concerns of this paper include:

1. The stress for nanoindentation is compressive, while the stress in the tensile test is tensile, how a compressive behavior can be used to predict the tensile behavior? 

2. It is recommended to add motivation in the abstract.

3. Line 73-76, references are not found in the manuscript.

4. Line 430-435, it is recommended to organize the information in a table.

5. Line 470-483, again, it is recommended to organize the information in a table. The author already made a table for the test information for the macroscale tests in Table 2, so, another table of microscale tests can be used to compare the information clearer.

6. Where is the conclusion section?

Author Response

Reviewer’s comments and our point-by-point answers

This paper investigates the correlation between indentation creep and the long-term tensile creep behavior among three polymer materials. Empirical models are fit to find the corresponding creep parameters.

Answer: We thank the reviewer for the evaluation and comments, which helped us to improve the quality of our manuscript. Point-by-point answers to the comments follow. All changes are marked with the red font in the revised manuscript.

My major concerns of this paper include:

  1. The stress for nanoindentation is compressive, while the stress in the tensile test is tensile, how a compressive behavior can be used to predict the tensile behavior?

Answer: The reviewer is right that the tensile and indentation experiments exhibit different types of deformation. This is due to different experiment geometry, different stress fields in the specimens, and other effects, which are discussed in detail in section 2.3. The same section also explains that the prediction of tensile creep from indentation creep is an open question, but it seems to be possible, because the two methods (i.e. long-term tensile and indentation experiments) measure basically the same property – resistance of the material to long-term loading.  In any case, we added one additional sentence at the end of the first paragraph of Theoretical background section so that this was clearer.

  1. It is recommended to add motivation in the abstract.

Answer: The motivation has already been included in the abstract of the original manuscripts, namely in the 2nd and 3rd sentence of the abstract, where the objectives of the study are specified. Perhaps there was a misunderstanding, because in this study the motivation and objectives are the same.

  1. Line 73-76, references are not found in the manuscript.

Answer: We thank the reviewer for careful reading. There were wrong links in the original version of the manuscript. The links have been corrected during the revision.

  1. Line 430-435, it is recommended to organize the information in a table.

Answer: We decided to list the basic properties of the three polymers briefly in the text of the Experimental section, because these are literature data and not our own measurements. Moreover, the table would show just a few values and occupy more space than the brief textual description.

  1. Line 470-483, again, it is recommended to organize the information in a table. The author already made a table for the test information for the macroscale tests in Table 2, so, another table of microscale tests can be used to compare the information clearer.

Answer: The reviewer is right that the loads in tensile creep experiments were summarized in the form of a table. The reason was that every polymer was tested at four different loads, and the loads were specific for each polymer, as explained in the text. Nevertheless, for indentation creep experiments we used just two loadings (50 gf and 200 gf) and the two loadings were the same for all three investigated polymers. Consequently, the table is not suitable here, as it would show just two identical values for all three polymers.

  1. Where is the conclusion section?

Answer: The conclusion section was added to the revised manuscript. The conclusion section is not obligatory in Materials journal, but we agree that a brief summary of the main results at the end of the text is suitable in this work.

Reviewer 2 Report

The paper studied the correlation of microscale indentation creep to macroscale creep behaviors of three of the most commonly used plastic with a broad range of physical properties. It is very useful for real life application of the polymers and to predict their macro creep behavior. The background and goals of the work was well laid out. The demonstration of the tests and results are clear and informative. The reviewer recommend this paper to be published with minor format editing. 

Author Response

Reviewer’s comments and our point-by-point answers

The paper studied the correlation of microscale indentation creep to macroscale creep behaviors of three of the most commonly used plastic with a broad range of physical properties. It is very useful for real life application of the polymers and to predict their macro creep behavior. The background and goals of the work was well laid out. The demonstration of the tests and results are clear and informative. The reviewer recommends this paper to be published with minor format editing.

Answer: We thank the reviewer for his/her positive evaluation. We re-checked the manuscript and made various minor corrections and improvements of the text as suggested.

Reviewer 3 Report

A few major and minor comments for the Authors before considering the present work for publication:   Major comments  
  1. At line 195 the definition of a generic time-dependent creep compliance is not correct. Please verify;
  2. Line 200-202: "The remaining EVP model parameters 200 (CvCi an τi) do not have any direct physical meaning and should be taken simply as constants of the model describing the time-dependent behavior of the material". I would rephrase the sentence because the EVP model parameters C_v, C_i, and tau_i should have physical meaning as follows: C_v = Inverse of zero-shear rate viscosity, C_i = compliance magnitude, tau_i = retardation time (time required for the extension of the spring to its equilibrium length while being retarded by the dashpot) of the ith KV element. Please comment;
  3. I got confused about the actual meaning of R2 (fit and all): Was the regression applied to all the experimental data sets for each system or to an average data set? Please explain; I would also suggest to make a little comment on the repeatability of the measurements to better understand the reason why 10 trials were made for instance;
  4. Considering the length of the present manuscript (+19 active pages) I would suggest to add a little paragraph for Conclusions to summarize all the key findings and introduce the reader on future works.
  Minor comments: i. Between line 73 and 77 three references are reported as "Error! Reference source not found". Please verify; ii. Line 97 - Figure 1c, light red curve. Please correct; iii. Line 98 - Figure 1c, medium and dark red curves. Please correct.

Author Response

Reviewer’s comments and our point-by-point answers

A few major and minor comments for the Authors before considering the present work for publication:

Answer: We thank the reviewer for the all comments, which helped us to improve our contribution. Point-by-point answers to the comments follow. All changes are marked with the red font in the revised manuscript.

Major comments  

At line 195 the definition of a generic time-dependent creep compliance is not correct. Please verify.

Answer: Corrected. Now it reads C_v = 1/eta.

Line 200-202: "The remaining EVP model parameters 200 (CvCi an τi) do not have any direct physical meaning and should be taken simply as constants of the model describing the time-dependent behavior of the material". I would rephrase the sentence because the EVP model parameters C_v, C_i, and tau_i should have physical meaning as follows: C_v = Inverse of zero-shear rate viscosity, C_i = compliance magnitude, tau_i = retardation time (time required for the extension of the spring to its equilibrium length while being retarded by the dashpot) of the ith KV element. Please comment.

Answer: Corrected as the reviewer suggested. The changed text is marked with the red font. Nevertheless, we kept the note that the above discussed quantities are just parameters of the model describing the time-dependent behavior, rather than real material constants. This is based on detailed discussion in the literature (references [6,33] in the revised manuscript).

I got confused about the actual meaning of R2 (fit and all): Was the regression applied to all the experimental data sets for each system or to an average data set? Please explain; I would also suggest to make a little comment on the repeatability of the measurements to better understand the reason why 10 trials were made for instance.

Answer to the first part of the comment: The exact meaning of R2(fit) and R2(all) was explained in an extra paragraph, which was added at the end of section 4.2. Briefly, R2(fit) was calculated for the data in the fitted interval (4-600s) in the case of long-term microindentation creep, while R2(all) was calculated for the data in the whole interval (4-3000s) in the case of long-term microindentation.

Answer to the second part of the comment: The reason why we measured indentation multiple times was explained in the first paragraph of section 4.1 of the revised manuscript. The inserted sentence is marked with red font. Briefly, multiple measurements were made in order to increase reliability and reproducibility of our measurements.

Considering the length of the present manuscript (+19 active pages) I would suggest to add a little paragraph for Conclusions to summarize all the key findings and introduce the reader on future works.

Answer: The conclusion was added to the revised manuscript as the reviewer suggested.

Minor comments:

  1. Between line 73 and 77 three references are reported as "Error! Reference source not found". Please verify;

Answer: The wrong references were corrected – we thank the reviewer for his/her careful check of our manuscript.

  1. Line 97 - Figure 1c, light red curve. Please correct;

Answer: The text was corrected so that it corresponded to the colors in the figure.

iii. Line 98 - Figure 1c, medium and dark red curves. Please correct.

Answer: The text was corrected so that it corresponded to the colors in the figure. 

Reviewer 4 Report

The article is very well-designed and written. The authors need to address the following minor queries.

1.       Abstract is not effective. Please re-write the abstract with quantitative results.

2.       (section Error! Reference source not found.), Please correct it and check the same for the entire manuscript.

3.       Authors are advised to add the SI units of the factors used in the equations. For ex. with length L (mm), width W (mm) and thickness T (mm).

4.       “primary creep (Figure 1c, ligt red curve).” Is it light or ligt red curve?

5.       “physics [23,28] and so we will just them briefly.” Not giving any meaning.

6.       “density (ISO 1183) 430 960 kg/m3” Correct the unit of density “3” must be superscripted. Check all.

7.       “Dumb-bell specimens (described in section 0)” Section 0?

8.       “For both methods (MH and MHI)” Initial mention the full form of MH and MHI, then use short form.

9.       For figures 9 and 10 add the error graph.

1Conclusion part is missing.

Author Response

Reviewer’s comments and our point-by-point answers

The article is very well-designed and written. The authors need to address the following minor queries.

Answer: We thank the reviewer for positive evaluation. Point-by-point answers to reviewer’s minor comments are given below. All changes in the revised manuscript are marked with red font.

  1. Abstract is not effective. Please re-write the abstract with quantitative results.

Answer: The reviewer is right that the manuscript describes all observed trends qualitatively. The reason is that the amount of quantitative data is really huge in this study. Therefore, we decided to describe the main trends in the abstract, to present the most important results in the main text, and to put all other results to the Appendixes and to the supplementary information file.

  1. Section Error! Reference source not found; please correct it and check the same for the entire manuscript.

Answer: The manuscript was re-checked and all wrong references were corrected.

  1. Authors are advised to add the SI units of the factors used in the equations. For ex. with length L (mm), width W (mm) and thickness T (mm).

Answer: The adding of units to all equations would make them less comprehensible and harder to read. Moreover, it is not a common practice. Nevertheless, we added the unit descriptions to the MCREEP package, which was used for all calculations in this study. This can be easily verified on www-site of the program (https://pypi.org/project/mcreep/).

  1. “primary creep (Figure 1c, light red curve).” Is it light or dark red curve?

Answer: It should be light red curve. The mistake was corrected. We thank the reviewer for careful check of the figure and text.

  1. “physics [23,28] and so we will just them briefly.” Not giving any meaning.

Answer: The sentence was rephrased. Now it reads: Various types of EVP models are covered in most textbooks of polymer physics [23,28] and so we will just revise them briefly.

  1. “density (ISO 1183) 430 960 kg/m3” Correct the unit of density “3” must be superscripted. Check all.

Answer: Re-checked and corrected wherever necessary.

  1. “Dumb-bell specimens (described in section 0)” Section 0?

Answer: We thank the reviewer for careful reading – the wrong reference was corrected.

  1. “For both methods (MH and MHI)” Initial mention the full form of MH and MHI, then use short form.

Answer: We did our best to minimize the re-definition of MH and MHI in the revised manuscript.

  1. For figures 9 and 10 add the error graph.

Answer: The error bars were added to both figures.

  1. Conclusion part is missing.

Answer: Conclusion section is not obligatory in Materials journal, but the reviewer (in accord with theother two reviewers) is right that in this case, some summary should be added. Therefore, we added a brief Conclusion section in the revised manuscript.

Reviewer 5 Report

The authors reviewed viscoelastic models and the tensile creep experiments for selected polymer materials. The experimental results were fitted against the models that provided by the software package the authors developed. The fitting results suggest that the models can fit well against the short-term creep experiments, while “long-term” creep experiments show significant differences between the models and experimental data.

Major point:

The referee believes that the authors should provide more argument that the classical viscoelastic models can be applied to describe the creep experiments, especially why a linear combination of a Maxwell model and multiple KV model were selected. On surface, these models can only fit very short-term data and long-term data fitted very poorly. On the other hand, as per the referee’s perspective, a more generalized fractional model usually gives a better fitting and prediction.

Minor point:

The source code of the package is unavailable. Upon the referee checking, the link on https://pypi.org/project/mcreep to project home is not working.

The authors should provide a correlation matrix of parameters, along with the fitting results. This will help readers and reviewers judge how much parameters are correlated (highly correlated parameters are less independent and more possible a sign of overfitting)

Author Response

Reviewer’s comments and our point-by-point answers

The authors reviewed viscoelastic models and the tensile creep experiments for selected polymer materials. The experimental results were fitted against the models that provided by the software package the authors developed. The fitting results suggest that the models can fit well against the short-term creep experiments, while “long-term” creep experiments show significant differences between the models and experimental data.

Answer: We thank the reviewer for the evaluation our work and the comments, which helped us to improve the quality of our manuscript. Point-by-point answers to reviewer’s minor comments are given below. All changes in the revised manuscript are marked with red font.

Major point:

The referee believes that the authors should provide more argument that the classical viscoelastic models can be applied to describe the creep experiments, especially why a linear combination of a Maxwell model and multiple KV model were selected. On surface, these models can only fit very short-term data and long-term data fitted very poorly. On the other hand, as per the referee’s perspective, a more generalized fractional model usually gives a better fitting and prediction.

Answer: The selected elasto-visco-plastic models were intensively tested and recommended for the indentation creep evaluation (references [6, 30-33] in the revised manuscript). They showed to be very good for description of both indentation and tensile creep within given time interval. Nevertheless, we have demonstrated that they were not very good for the prediction of the creep beyond the fitted interval, while the simple empirical power law was very good for this. The reviewer is right that even more sophisticated models (such as generalized fractional models) might yield better creep predictions. This information was added (together with appropriate references) to the Discussion section (section 5.4, the last item, marked with red font).

Minor point:

The source code of the package is unavailable. Upon the referee checking, the link on https://pypi.org/project/mcreep to project home is not working.

Answer: The www-site was updated and the links were corrected.

The authors should provide a correlation matrix of parameters, along with the fitting results. This will help readers and reviewers judge how much parameters are correlated (highly correlated parameters are less independent and more possible a sign of overfitting)

Answer: We calculated the covariance matrices for selected cases, confirming that the increasing number of parameters in EVP models increases the covariance (i.e. decreases the stability of the fit and mutual independence of the regression parameters). The covariance matrices were added to the supplementary material.